# Anomaly Detection using Cascade Variational Autoencoder Coupled with Zero Shot Learning

**Gokul Ramasamy**                                    GRAMASAM@ASU.EDU
*Arizona State University, Tempe, AZ, USA*

**Bhavik N. Patel**                                    PATEL.BHAVIK@MAYO.EDU
*Mayo Clinic, Phoenix, AZ, USA*

**Imon Banerjee**                                    BANERJEE.IMON@MAYO.EDU
*Mayo Clinic, Phoenix, AZ, USA*

## Abstract

Detection of anomalies before they are included in the downstream diagnosis/prognosis models is an important criterion for maintaining the medical AI imaging model performance across internal and external datasets. However, the core challenges are: (i) given the infinite variations of possible anomaly, curation of training data is in-feasible; (ii) making assumptions about the types of anomalies are often hypothetical. We propose an unsupervised anomaly detection model using a cascade variational autoencoder coupled with a zero-shot learning (ZSL) network that maps the latent vectors to semantic attribute space. We present the performance of the proposed model on two different use cases – skin images and chest radiographs and also compare against the same class of state-of-the art generative OOD detection models.

**Keywords:** Anomaly detection, OOD, Zero-shot learning, medical imaging

## 1. Introduction

The performance of deep learning models, especially supervised learning has been shown to be on par with health-care professionals in multiple applications (Liu et al., 2019; Rajpurkar et al., 2017). Despite the high performance, the safety and reliability of these models is questioned, as often the models fail to retain the same performance on the unseen external dataset. Inaccurate predictions can cause a catastrophic consequence for the patients when applied in clinical diagnosis and prognosis. One major reason for the performance drop on external dataset is that supervised learning models operate under closed-world assumption (Fei and Liu, 2016) i.e., during inference, the models can handle only samples which contains exactly similar pattern to the data that the model has been trained on. But that's not often the case after deployment in the real world. As a simple example, the model could be trained with a regular chest radiograph dataset and during the testing, it receives a chest radiograph dataset with a mix of regular and high contrast CLAHE images, and the model fails terribly.

Another important aspect of improving the diagnostic performance of supervised AI models is the need for large amounts of high quality training data. But challenge with curating high-quality medical data is that the datasets from different institutions can be heterogeneous with distribution shifts (Cao et al., 2020).

*OOD* data, also called *anomaly/ outlier*, usually refers to data that shows dissimilarity from the training data distribution and often AI models fail to retain performance on

the OOD data. A successful open-world deployment of an AI model with OOD detection should be sensitive to unseen classes and distribution-shifted samples and also be resilient to potential adversarial attacks (Sehwag et al., 2019). To train a OOD detector with only in-distribution (ID) data available, learning high-quality "normality" features is the fundamental step to identify the OOD samples during inference. We designed a novel OOD architecture by combining generative and zero-shot learning model and performed a comparartive analysis against the state-of-the-art GAN based OOD - f-AnoGAN (Schlegl et al., 2019) and autoencoder based CVAE model (CVAD) (Guo et al., 2021).

## 2. Methodology

Our proposed architecture has two components - CVAE (Cascade Variational Autoencoder) and ZSL (Zero-Shot Learning) network. CVAE is similar to the generator of CVAD (Guo et al., 2021). In contrast to Vanilla VAE, a cascaded architecture provides high-quality reconstructions and better latent representations. The CVAE has two branches of VAE. The primary branch comprising of $E_1$ and $D_1$ as the encoder and decoder, while the secondary branch comprises of $E_2$ and $D_2$ as the encoder and decoder. The input to the secondary branch is the concatenated feature vector from $E_{11}$ and $D_{11}$ to improve the quality of the generated images. The CVAE network is trained with KL Divergence loss and Mean-

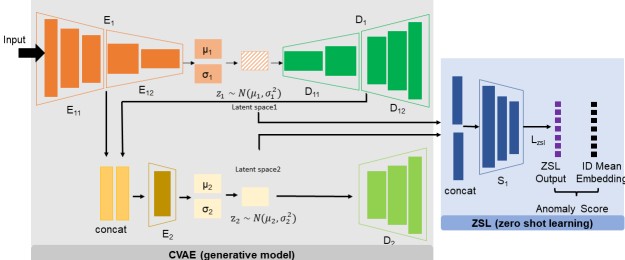

Figure 1: Proposed Architecture of OOD with CVAE and ZSL.

Squared Error (MSE) loss which is used as the reconstruction loss. The ZSL network $S_1$ is designed with three linear layers and a sigmoid activation at the end. The concatenated vector of the latent vectors from $E_1$ and $E_2$ of size 2560 (2048, 512) is the input to the ZSL network. The size of the output layer is determined by the number of semantic features used in the particular use-case. Ideally, the distance between the predicted attributes and the ground truth attributes would serve as the anomaly score. But here the external dataset is not as extensively labelled with the chosen semantic attributes. Hence, the semantic outputs of the trained ZSL model on the internal ID dataset is averaged to get a mean ID embedding which would serve as a representative embedding for ID data. The predicted external ID data embedding are closer to the representative embedding whereas the predicted external OOD data embedding is not. The Euclidean distance between the predicted embedding and the representative ID embedding is used to model this variation and serves as the anomaly score. The mean distance between the representative ID embedding and the internal ID data serves as a threshold to classify the external data as ID and OOD.

## 3. Experiments and Results

The proposed architecture is validated on two distinct medical problems and used four publicly available imaging datasets: (i)*Dermatology Images* - ISIC Dataset (internal) and Fitzpatrick17k (external); (ii)*Chest Radiographs* - CheXpert Dataset (internal) and UNIFESP Dataset (external). For our use cases, three distinct out-of-distribution categories are identified - *Type 1 (Domain Shift)* - Data samples totally unrelated to the task at hand; *Type 2 (Quality drift)* - Data samples that were acquired incorrectly; *Type 3 (Interclass OOD)* - Data samples that are unseen due to selection bias. Within the scope of this work, the internal test set is tested predominantly on type 3 (interclass OOD), while the external data is tested on a set of images comprising of all the use cases (type 1-3). The CVAE network coupled with ZSL was trained on ISIC 2020 with 21 different semantic attributes (e.g. anatomic region, skin color) and CheXpert with 23 different attributes (e.g. support device, fracture, gender). The number of training and validation images for pre-training CVAE and training the entire network are presented in Table 1.

Table 1: Training and Validation data statistics for CVAE pre-train and ZSL network train

|  | Dataset | Train | Validation |
|---|---|---|---|
| **CVAE pre-train** | ISIC 2019 | 11193 | 947 |
|  | CheXpert | 9900 | 1100 |
| **CVAE+ZSL train** | ISIC 2020 | 4358 | 600 |
|  | CheXpert | 5000 | 1000 |

The performance of our proposed architecture (CVAE+ZSL) are tested against f-AnoGAN (Schlegl et al., 2017) and CVAD (Guo et al., 2021) (Table 2). The threshold was chosen as $mean + 0.5 * std$ by observing the mean and the standard deviations of the anomaly scores for AnoGAN and CVAD, and the Euclidean distance for our methodology. From our experimental results, it can be observed that the proposed CVAE+ZSL model outperformed both AnoGAN and CVAD on the external unseen data with significant *data shift*.

Table 2: Comparative analysis of AnoGAN, CVAD and CVAE+ZSL

| Models | AnoGAN | | | CVAD | | | CVAD +ZSL | | |
|---|---|---|---|---|---|---|---|---|---|
| Dataset | Acc ↑ | TPR ↑ | FPR ↓ | Acc ↑ | TPR ↑ | FPR ↓ | Acc ↑ | TPR ↑ | FPR ↓ |
| **ISIC ID + OOD** (internal) | 0.575 (0.561, 0.577) | 0.305 (0.289, 0.307) | 0.155 (0.146, 0.160) | **0.70** (**0.688, 0.703**) | 0.72 (0.702, 0.722) | 0.32 (0.311, 0.331) | 0.5725 (0.560, 0.577) | 0.35 (0.334, 0.354) | 0.205 (0.196, 0.212) |
| **Fitzpatrick17k ID + OOD** (external) | 0.49 (0.488, 0.504) | 1.0 | 1.0 | 0.334 (0.325, 0.338) | 0.36 (0.353, 0.372) | 0.691 (0.689, 0.708) | **0.6511** (**0.643, 0.658**) | 0.555 (0.543, 0.566) | 0.256 (0.246, 0.264) |
| **CheXpert ID + OOD** (internal) | **0.7707** (**0.768, 0.776**) | 0.9172 (0.918, 0.932) | 0.2433 (0.238, 0.248) | 0.7076 (0.703, 0.712) | 0.2966 (0.284, 0.305) | 0.2522 (0.252, 0.261) | 0.6778 (0.663, 0.674) | 0.4896 (0.473, 0.506) | 0.304 (0.298, 0.305) |
| **UNIFESP ID + OOD** (external) | 0.6987 (0.681, 0.699) | 1.0 | 0.411 (0.411, 0.431) | 0.4226 (0.412, 0.426) | 0.25 (0.229, 0.260) | 0.512 (0.509, 0.525) | **0.7112** (**0.701, 0.716**) | 0.5468 (0.535, 0.564) | 0.2286 (0.224, 0.239) |

## 4. Conclusion

The proposed work designed an unsupervised anomaly detection model using a cascade coupled with zero-shot learning network maps the latent vectors to semantic attributes. With the inclusion of semantic space, the proposed architecture generalizes well and has a better performance on the unseen external dataset when compared against the same class of state-of-the art models.

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
