# OpenReview forum: "Anomaly Detection using Cascade Variational Autoencoder Coupled with Zero Shot Learning "
_MIDL.io/2023/Short_Paper_Track — MIDL 2023 Short paper track Poster_

### Official Review · Reviewer_zVMc · 2023-04-20
**Paper 94 review**

**Rating:** 8
**Confidence:** 5

**Review:**

This is a novel approach to anomaly detection, by using zero shot learning and cascaded VAE. The results are certainly promising, and definitely deserves to b showcased

---

### Official Review · Reviewer_h1WC · 2023-04-26
**Review of Anomaly Detection using Cascade Variational Autoencoder Coupled with Zero Shot Learning**

**Rating:** 6
**Confidence:** 4

**Review:**

This paper presents a method for anomaly detection (a.k.a., out-of-distribution (OOD) detection) using a cascaded variational autoencoder (CVAE) concatenated with a zero shot learning network. The method is tested on four different medical imaging datasets.

* The model seems to be well designed, and it performs relatively well in the experiments.

* The motivation for the cascaded VAE is not clear, in other words, what does the cascading add to this?

* It wasn't clear to me how you go from the anomaly score out of the zero shot network to get a final classification as an anomaly.

* The results are mixed, with the proposed method "winning" half of the time, and competing methods "winning" the other half.

* There are a lot of OOD detection methods in the literature. The authors only compare against one fairly old one (AnoGAN, 2017) and there own previous method (CVAE without ZSL). Is this the best comparison to state-of-the-art?